# The Role of BDNF and TrkB in the Central Control of Energy and Glucose Balance: An Update

**DOI:** 10.3390/biom14040424

**Published:** 2024-03-31

**Authors:** Theresa Harvey, Maribel Rios

**Affiliations:** 1Graduate Program in Neuroscience, Graduate School of Biomedical Sciences, Tufts University School of Medicine, Boston, MA 02111, USA; theresa.harvey@tufts.edu; 2Department of Neuroscience, Tufts University School of Medicine, 136 Harrison Avenue, Boston, MA 02111, USA

**Keywords:** BDNF, TrkB, hypothalamus, plasticity, energy, balance, glucose, obesity

## Abstract

The global rise in obesity and related health issues, such as type 2 diabetes and cardiovascular disease, is alarming. Gaining a deeper insight into the central neural pathways and mechanisms that regulate energy and glucose homeostasis is crucial for developing effective interventions to combat this debilitating condition. A significant body of evidence from studies in humans and rodents indicates that brain-derived neurotrophic factor (BDNF) signaling plays a key role in regulating feeding, energy expenditure, and glycemic control. BDNF is a highly conserved neurotrophin that signals via the tropomyosin-related kinase B (TrkB) receptor to facilitate neuronal survival, differentiation, and synaptic plasticity and function. Recent studies have shed light on the mechanisms through which BDNF influences energy and glucose balance. This review will cover our current understanding of the brain regions, neural circuits, and cellular and molecular mechanisms underlying the metabolic actions of BDNF and TrkB.

## 1. Introduction

Brain derived-neurotrophic factor (BDNF) is a highly conserved member of the family of neurotrophins, playing chief roles in neuronal survival and differentiation during development and synaptic plasticity and neurotransmission in the adult brain. It is expressed primarily by excitatory neurons and signals through the p75 receptor as well as the tropomyosin-related kinase B (TrkB) receptor, which activates the phospholipase C gamma (PLCγ), the mitogen-activated protein kinase (MAPK), and the phosphatidylinositol-3 kinase (PI3K) intracellular signaling cascades [1]. In addition to full-length TrkB, the *Ntrk2* locus also generates truncated forms of the receptor, including TrkB.T1, through alternative splicing. TrkB.T1 lacks the intracellular tyrosine kinase domain and binds BDNF with the same affinity as TrkB. Historically, TrkB.T1 has been described as an antagonist of TrkB, serving as a dominant negative receptor as well as a BDNF scavenger [2]. It is now recognized that in addition to those functions, TrkB.T1 signals independently primarily in astrocytes to regulate astrocytic calcium entry and morphology via regulation of Rho GTPase activity and the actin cytoskeleton [3].

BDNF was first implicated in the central control of energy balance by the observation that chronic intracerebroventricular (ICV) administration of this neurotrophin in rats reduced feeding and body weight [4,5,6]. Subsequent genetic studies showed that BDNF^+/−^ mutant mice exhibited hyperphagic behavior and elevated body weights, further indicating the importance of BDNF in energy balance regulation [7]. Similarly, TrkB hypomorphic mice, expressing TrkB at approximately 25% of normal levels, displayed severe obesity, hyperphagia, and hyperdipsia, and these conditions were exacerbated when the mutants were fed a moderate-fat diet [8]. Following studies involving the examination of mice with central depletion of BDNF, it became evident that actions of this neurotrophin in the brain are critical to achieving energy balance control. These BDNF^2L/2LCk-cre^ mutant mice experience an extensive reduction in BDNF in Ca^2+^/calmodulin-dependent protein kinase II-expressing neurons throughout the postnatal brain, with the exception of the cerebellum [9]. BDNF^2L/2LCk-Cre^ mice fed a chow diet exhibit mature onset obesity, with female and male mutants weighing 150 and 80% more than sex-matched controls, respectively. They display hyperphagic behavior when fed a chow or a high-fat diet (HFD) and manifest other aspects of the metabolic syndrome, including leptin and insulin resistance, dyslipidemia, and hyperglycemia [9]. It is worth noting that BDNF^2L/2LCk-Cre^ mice also show increased locomotor activity, likely offsetting some of the excessive weight gain associated with increased food consumption.

Following those early studies, ensuing investigations have provided a more granular understanding of the actions of BDNF and TrkB that facilitate energy balance and glycemic control. Considering that mutations in *Bdnf* and *Ntrk2* are also associated with hyperphagia and obesity in humans, these advances in knowledge are significant and necessary. For example, the single-nucleotide polymorphism Val66Met within the *Bdnf* gene is highly prevalent in the human population and is associated with obesity susceptibility and type-2 diabetes in adults and children [10,11,12]. Furthermore, individuals with WAGR syndrome and related BDNF haploinsufficiency exhibit childhood-onset obesity and lower BDNF serum levels [13]. Patients with a heterozygous missense mutation, Y722C in *Ntrk2*, have severe hyperphagia and obesity, diminished learning and memory, and impaired detection of painful stimuli [14]. This Y722C mutation, occurring at one of the key tyrosine residues within the tyrosine kinase domain, prevents receptor phosphorylation and downstream signaling via MAPK [15]. Additionally, hypothalamic BDNF and TrkB expression is significantly downregulated in Prader–Willi syndrome patients, who are susceptible to hyperphagic obesity [16].

Energy and glucose homeostasis are regulated by complex neural circuits residing primarily within the hypothalamus and the hindbrain. In this review, we discuss the circuitry and the cellular and molecular mechanisms within those regions underlying the metabolic effects of BDNF and TrkB. Although the discussion emphasizes the hypothalamic control of metabolic function by BDNF/TrkB signaling, findings indicating significant roles in other brain areas are also described.

## 2. BDNF Signaling and Hypothalamic Control of Energy and Glucose Balance

The hypothalamus plays an essential role in physiological homeostasis, including the regulation of energy and glucose balance. Hypothalamic neural circuits integrate hormonal and nutrient signals informing changes in body energy state and, in response, mediate adaptive changes in food intake, energy utilization, and glucose mobilization [17,18]. In a state of positive energy balance, elevated levels of nutrients and appetite-suppressing peptides and hormones, including the adipokine leptin, act on hypothalamic cells to heighten the anorexigenic tone and increase energy expenditure. Conversely, when energy stores are depleted, increased gastric secretion of the orexigenic hormone ghrelin leads to the activation of hypothalamic signaling cascades that stimulate eating and reduce energy expenditure.

Several interconnected hypothalamic nuclei influence metabolic function, including the arcuate nucleus (Arc), paraventricular nucleus (PVN), ventromedial hypothalamus (VMH), dorsomedial hypothalamus (DMH), and lateral hypothalamus (LH). BDNF is produced in the VMH, PVN, DMH, and LH, whereas TrkB is present in the PVN, DMH, LH, VMH, and Arc [8,19,20,21]. Interestingly, subcellular localization of a distinct population of *Bdnf* transcripts in the hypothalamus influences energy homeostasis. Use of alternative polyadenylation sites produces *Bdnf* mRNAs containing either short (~0.4 kb) or long (~2.9 kb) 3′ untranslated regions (3′UTR) and with the same coding sequence [22]. These unique 3′UTRs direct specific subcellular localization and sites of BDNF translation. Short 3′UTR *Bdnf* transcripts localize to neuronal cell bodies, whereas long 3′UTR *Bdnf* transcripts reside in neuronal dendrites in the cortex and hippocampus, where they serve as templates for local BDNF protein synthesis [23]. Liao et al. showed that this subcellular distribution of *Bdnf* transcripts is also evident in the hypothalamus and that leptin and insulin induce local translation of long 3′UTR *Bdnf* in dendrites of cultured hypothalamic neurons [24].

The functional significance of this differential localization of *Bdnf* transcript forms was explored using knock-in mice lacking dendritically targeted long 3′UTR *Bdnf* mRNA (*Bdnf*^klox/klox^) due to an insertion of three tandem simian virus 40 (SV40) polyadenylation signals into the genomic sequence encoding the long *Bdnf* 3′ UTR [24]. These animals exhibited severe obesity, with female and male mutants weighing 171% and 90% more, respectively, than sex-matched controls. Increases in body weight were associated with hyperphagia, hyperglycemia, hyperleptinemia, and increased adiposity. Leptin administration to *Bdnf^klox/klox^* mice did not elicit a reduction in food intake, as observed in wild-type mice, indicating that long 3′UTR *Bdnf* mRNA is necessary for leptin’s anorexigenic effect [24]. Notably, metabolic alterations in *Bdnf^klox/klox^* mice were reversed by viral delivery of long 3′UTR *Bdnf* mRNA but not short 3′UTR *Bdnf* mRNA to the hypothalamus.

These findings highlight the physiological relevance of alternative *Bdnf* transcript species and how their subcellular localization in the hypothalamus regulates energy balance. They also suggest that leptin induces dendritic translation of 3′UTR *Bdnf* mRNA to rapidly and persistently strengthen or weaken hypothalamic synapses to accommodate the energy demands of the animal [23,24]. It is now clear that BDNF and TrkB act in several hypothalamic nuclei to influence metabolic function (Figure 1). Below, we describe each of these energy balance-regulating centers in more detail and the effects of BDNF/TrkB signaling in those regions.

### 2.1. Arcuate Nucleus

Because the Arc has the capacity to sense circulating hormonal and nutrient signals informing the animal’s energy status due to its proximity to fenestrated capillaries at the base of the hypothalamus [25], it functions as a key node in the regulation of energy and glucose homeostasis. It houses two distinct types of neurons that produce proopiomelanocortin (POMC) or agouti-related protein and NPY (AgRP/NPY), which inhibit and drive eating, respectively [26,27,28,29,30,31,32]. POMC is a precursor polypeptide for α-melanocyte stimulating hormone (α-MSH), the anorexigenic ligand of melanocortin receptor 4 (MC4-R) [17,33]. AgRP, for its part, acts as an inverse agonist of MC4-R, counteracting the anorexigenic effects of α-MSH and increasing the drive to eat. AgRP neurons also produce and release NPY and GABA to elevate the orexigenic tone and reduce energy utilization [34]. Leptin, along with other metabolic cues, acts directly on these cell populations in the Arc, which in turn transmit these signals to other hypothalamic areas, including dense projections to the PVN, to modulate neural pathways that regulate food intake, energy expenditure, and glucose homeostasis [17].

Even though BDNF is not expressed in the Arc [8], cells in this region produce TrkB, including astrocytes and a small fraction of POMC^+^ (16.5%) and NPY^+^ (7.8%) neurons [35]. Work by Liao et al. (2015) indicates that intact BDNF signaling is requisite for the development of axonal projections of a subset of these cells. In support, *Bdnf^klox/klox^* mice lacking 3′UTR *Bdnf* mRNA exhibit a reduction in the number of Arc AgRP axons targeting the lateral part of the PVN as well as the number of Arc α-MSH-containing projections to the dorsal and ventral subregions of the DMH [35]. The synaptic organization of AgRP and POMC neurons in the Arc is also altered in *Bdnf^klox/klox^* mice. Accordingly, a reduction and an increase in the density of excitatory synaptic inputs onto NYP and POMC cell bodies, respectively, was observed in mutant mice [36]. Because these animals exhibit hyperphagia and severe obesity, this synaptic reorganization might represent a compensatory homeostatic mechanism to restore energy balance control [36].

There is also evidence suggesting that BDNF/TrkB signaling in the Arc of adult animals is involved in the regulation of energy balance. For example, leptin treatment in wild-type mice results in elevated pSTAT3 levels in the Arc in a subset of TrkB^+^ neurons [35]. Additionally, activity of these cells was elevated during refeeding following a fasting period compared to normally fed mice. Future investigations should interrogate the necessity of TrkB signaling in the Arc in the regulation of metabolic function.

### 2.2. Paraventricular Nucleus

The PVN contains a diverse pool of neurons involved in the regulation of feeding, including those expressing MC4R and oxytocin [27,37,38]. It serves as a pivotal metabolic hub receiving extensive intrahypothalamic input, including dense POMC and AgRP fibers from the Arc, and projecting to feeding-regulating regions of the hindbrain, including the parabrachial nucleus [39,40]. Functioning as a critical autonomic control center, it also influences sympathetic tone in metabolic organs to regulate thermogenesis and glucose homeostasis [41]. Early findings showing that BDNF delivery to the PVN of rodents increased energy expenditure and reduced feeding implicated this hypothalamic region in the metabolic actions of BDNF [42,43,44]. BDNF-containing neurons in the PVN are distinct from those expressing MC4R or oxytocin, and BDNF in this region is essential for the regulation of energy balance. In support, mice (Sim1-Cre:Bdnf ^lox/lox^) with BDNF depletion in Sim1^+^ cells, which blanket the PVN and represent several molecularly defined neuronal subpopulations, exhibit hyperphagia, obesity, glucose intolerance, and hyperinsulinemia compared to control littermates fed a chow diet [36]. Male Sim1-Cre:Bdnf^lox/lox^ mutants were 41% heavier than sex-matched controls, whereas females exhibited a 76% increase in body weight at twenty weeks of age. Male and female Sim1-Cre:Bdnf^lox/lox^ mutant mice also display reduced oxygen consumption and locomotor activity during the dark cycle as well as deficient adaptive thermogenesis in brown adipose tissue (BAT), indicating reduced energy expenditure [36].

Because Sim-Cre targets brain regions other than the PVN, including the amygdala [45], follow-up studies focused on selectively knocking down BDNF expression in the adult PVN via viral delivery of Cre recombinase to this area in floxed BDNF mice. This manipulation resulted in increased energy intake as well as dramatic levels of obesity [36]. Interestingly, these effects were more severe than those observed in Sim1-Cre:Bdnf^lox/lox^ mutant mice, suggesting that compensatory mechanisms partially protect against metabolic dysfunction when BDNF is depleted from the PVN early in development. Another noteworthy observation made by this group is that BDNF^+^ neurons in the anterior PVN play a more prominent role in suppressing feeding, whereas those residing in the medial and posterior PVN are connected to brown adipose tissue (BAT) via a polysynaptic circuit regulating sympathetic outflow and thermogenesis [36]. PVH^BDNF^ neurons, which receive inputs from POMC and AgRP neurons in the Arc, have also been implicated in the positive effects of leptin regulating sympathetic innervation of adipose tissue. Accordingly, mice with selective BDNF knockdown in PVN exhibit decreased innervation of thermogenic inguinal white adipose tissue (iWAT) and BAT [46]. Furthermore, selective ablation of PVH^BDNF^ neurons in *ob/ob* mice lacking leptin blunted the effects of exogenous leptin treatment, increasing the sympathetic innervations of iWAT and BAT in these mice and restoring BAT temperature to control levels. In total, the data indicate that PVH^BDNF^ neurons act directly downstream of AgRP and POMC neurons in the Arc to mediate the effects of leptin on the sympathetic innervation of WAT and BAT [46].

In addition to projections from the Arc, PVH^BDNF^ neurons receive dense inputs from the VMH, the DMH, and the lateral parabrachial nucleus [47]. Chemogenetic studies further attest to the involvement of these cells in metabolic control, as their acute activation decreases nocturnal and fasting-induced food intake and increases thermogenesis, locomotor activity, and energy expenditure [48]. Interestingly, some of the effects elicited by activation of these cells are sexually dimorphic. Accordingly, acute activation of PVN^BDNF^ neurons resulted in a sustained increase in the respiratory exchange ratio (RER), a surrogate for substrate utilization as fuel, following an initial RER decrease in females but not in males. Furthermore, chronic activation of PVN^BDNF^ neurons in females did not affect body weight but instead increased fat while decreasing lean mass. The results suggest that chronic stimulation of PVN^BDNF^ neurons promotes long-term storage of fatty acids to compensate for elevated energy expenditure. PVN^BDNF^ cells mediate their effects, at least in part, via expression of retinoic acid-induced 1 (RAI1), which regulates expression of BDNF [49]. Accordingly, selective deletion of *Rai1* in BDNF^+^ neurons results in increased body weight, glucose intolerance, and decreased intrinsic excitability of PVN^BDNF^ neurons. This is an interesting finding considering that 40% of individuals with the disorder Smith–Magenis syndrome caused by interstitial deletions of the 17p11.2 chromosomal region containing *Rai1* are overweight [50,51].

TrkB^+^ neurons also localize to the PVN and are mostly distinct from those containing BDNF or MC4R. Indeed, only 5% and 3% of TrkB-containing neurons in the PVN express BDNF and MC4R, respectively [52]. These cells prominently participate in the control of energy homeostasis, as indicated by the increased food intake, reduced locomotor activity and energy expenditure, and severe obesity observed in mice depleted of TrkB throughout the PVN (Sim1-Cre*:Ntrk2^lox/lox^*) [52]. Similarly, selective depletion of TrkB in the adult PVN elicits obesity, and this effect is primarily linked to excessive food consumption, as energy expenditure is not affected [52]. Finally, whereas chemogenetic inhibition of PVN^TrkB^ neurons increased food intake during the light and dark cycles, their activation significantly reduced food consumption during the dark cycle or following an overnight fast. Because MC4R expression was normal following TrkB depletion in the PVN, it is unlikely that deficits in melanocortin signaling contribute to energy balance dysregulation. Moreover, selective depletion of TrkB in oxytocin-containing neurons had no effects on body-weight control [52], ruling out this cell population as an essential target of the anorexigenic actions of BDNF-TrkB signaling.

Viral tracing studies indicate that PVN^TrkB^ neurons project to the VMH, the medial eminence (ME), the lateral parabrachial nucleus (LPBN), and the nucleus tractus solitarious (NTS) [52]. Deleting *Ntrk2* in PVN^TrkB^ neurons that project to the VMH (PVN^TrkB→VMH^ neurons) or to the LPBN (PVH^TrkB→LPBN^ neurons) induces hyperphagia and obesity, highlighting that these TrkB neuronal projections are critical for suppressing appetite [52]. In total, the evidence indicates that the PVN is a critical source of BDNF for the regulation energy homeostasis. Furthermore, it shows that PVN^BDNF^ and PVN^TrkB^ cells are distinct neuronal populations with overlapping but not identical effects on energy balance control.

### 2.3. Ventromedial Hypothalamus

The VMH is the hypothalamic region with the highest content of BDNF mRNA, and the expression of this neurotrophin here is positively regulated by leptin [8,53]. It encompasses a diverse array of neurons, predominantly glutamatergic, that play key roles promoting satiety, increasing energy expenditure, and mediating glucose homeostasis via regulation of autonomic nervous system tone onto metabolic organs [54,55,56,57]. The finding that the expression of both BDNF and TrkB transcripts is sensitive to caloric status in the VMH provided the first indication that this hypothalamic region is a key substrate for the metabolic effects of this neurotrophin [8,21,58]. Whereas fasting resulted in a 58% reduction in BDNF mRNA levels in the VMH of wild-type mice, administration of glucose rapidly (30 min) restored BDNF transcripts in this region in fasted mice [21]. Glucose administration in fasted animals also elicited a robust increase in TrkB mRNA levels in the VMH. Reduced BDNF synthesis in the VMH was also reported in mice lacking steroidogenic factor 1 (SF-1), a transcription factor exclusively expressed in the VMH within the brain, which is requisite for terminal differentiation of VMH neurons and the regulation of energy and glucose balance [58].

A functional link between VMH BDNF and metabolic health was first indicated by studies showing that selective BDNF delivery to this region in rats diminished food intake while elevating energy expenditure, consistent with the idea that BDNF signaling promotes negative energy balance [59,60]. To interrogate the necessity of intact BDNF function in this hypothalamic nucleus, Unger et al. examined the effect of depleting it in the VMH of adult mice [21]. These studies also addressed a persistent question in the field: whether the hyperphagia and obesity exhibited by developmental rodent models of BDNF or TrkB deficiency were related to hardwired developmental alterations in the feeding circuitry or perturbations of important roles played in those circuits in adulthood. BDNF depletion in the adult VMH elicited increased weight gain, hyperphagia, hyperglycemia, hyperinsulinemia, and hyperleptinemia without affecting locomotor activity. Moreover, pair-feeding these BDNF mutants to the consumption levels of control mice normalized their body weights without affecting their core body temperature, indicating that impaired feeding regulation and not alterations in energy expenditure primarily contributed to elevations in body weight [21]. Although these studies did not rule out significant roles during development, they unequivocally demonstrated that intact BDNF signaling in the mature brain is required for metabolic control and that the VMH is an important cellular component mediating these effects. In contrast to these findings, developmentally depleting BDNF in SF1 neurons, which reduces BDNF content in the VMH by 50%, did not affect body weight but compromised the counterregulatory response to hypoglycemia, as indicated by persistent lower levels of glucose and impaired glucagon release following insulin-induced hypoglycemia [61]. Discrepancies between these two BDNF mutant models might be rooted in the adult versus developmental period of BDNF depletion in the VMH and that in the model utilized by Unger et al., BDNF depletion was not limited to SF1 neurons but rather occurred broadly in the VMH [21]. Therefore, it is possible that targeting *Bdnf* during development in the VMH invokes compensatory mechanisms that preserve energy balance control or, alternatively, that neuronal populations other than SF1 neurons are the critical source of BDNF regulating feeding and body weight.

The *Bdnf* gene contains several promoters preceding untranslated exons, each of which splices to a single coding exon to generate several transcript isoforms encoding for the same BDNF protein [22]. This gene structure and alternative promoter use provides a mechanism to regulate BDNF expression in a tissue-specific manner. *Bdnf* transcripts driven by promoter 1 are enriched in the VMH [21,58,62]. More recently, Chu et al. interrogated the effect of selectively disrupting *Bdnf* promoter 1 or 2 (*Bdnf-e1^−/−^* and *Bdnf-e2^−/−^*). *Bdnf-e1^−/−^* and *Bdnf-e2^−/−^* mutant mice weighed significantly more than wild-type littermates, and even though *Bdnf-e1^−/−^* mice exhibited normal food intake when group housed, they became hyperphagic following long-term social isolation or HFD administration [63]. On the other hand, *Bdnf-e2^−/−^* mice exhibited hyperphagia and obesity but no deficits in BAT thermogenesis. Notably, *Bdnf-e2^−/−^* mice with selective delivery of AAV-*e2*-BDNF to the VHM to restore *Bdnf*-e2 transcripts rescued body weight, lean and fat mass, and food intake to levels comparable to those of wild-type controls by 4 weeks post injection [63]. These studies suggest that selective activation of *Bdnf* promoters 1 and 2 in the VMH is critical for the regulation of energy homeostasis. Furthermore, studies from this group revealing that *Bdnf*-e1-expressing cells in the VMH also contain TrkB and that TrkB depletion in the VMH phenocopies the effects observed in *Bdnf-e1^−/−^* mice suggest that *Bdnf*-e1 might be acting, at least in part, via autocrine mechanisms.

A clearer understanding of the cellular mechanisms underlying the metabolic effects of BDNF in the VMH has emerged recently. Key among those findings is that the activity of VMH neurons is regulated by caloric status in a BDNF-dependent manner. Accordingly, we discovered that fasting triggers a significant reduction in the activity of VMH neurons compared to the fed state in wild-type mice [64]. These findings are consistent with previous studies indicating that VMH neurons are anorexigenic and thus, expected to be less active in a state of negative energy balance. Importantly, the firing rate of VMH neurons in fed BDNF^2L/2L:CK-Cre^ mice was significantly reduced compared to fed wild-type controls, was similar to that of fasted controls, and was not affected by caloric status. In other words, VMH neurons are in a persistent fasted level of hypoactivity in the absence of BDNF, increasing the drive to eat.

Changes in the excitatory drive but not in intrinsic excitability underlie energy status-driven changes in VMH neuronal activity. Accordingly, the frequency of spontaneous excitatory post-synaptic currents (sEPSCs) in these cells is increased in the fed state, when BDNF levels are higher, without changes in intrinsic excitability [64]. Notably, fed BDNF^2L/2L:CK-Cre^ mice displayed a significant rightward shift of the sEPSC interevent interval compared to fed control animals, indicating that excitatory events are less frequent in the absence of BDNF. Attesting to the sufficiency of BDNF driving this caloric state-associated plasticity, ICV delivery of BDNF in fasted wild-type mice fully and rapidly restored the frequency of sEPSCs to the levels observed in fed mice [64]. Synaptic remodeling contributes, at least in part, to the observed changes in excitatory tone in the VMH. Accordingly, fasting in wild-type mice significantly decreased the density of excitatory synapses in the dorsomedial and central (dm/cVMH) VMH, as marked by co-localization of the presynaptic and postsynaptic markers vGlut2 and PSD95, respectively. Moreover, fed BDNF^2L/2L:CK-Cre^ mice exhibited a reduced number of excitatory synapses in the dm/cVMH compared to fed wild-type mice [64].

Recent investigations indicate that astrocytic synaptic glutamate clearance, which greatly influences post-synaptic glutamate availability and neuronal excitability, also varies according to energy state in a BDNF-dependent manner in the VMH. It is now recognized that in addition to providing structural and metabolic support to neurons, astrocytes actively participate in neurotransmission via several mechanisms [65,66,67]. For example, extracellular glutamate is primarily removed by the astrocytic excitatory amino acid transporters (EAATs) GLT-1 and GLAST, limiting the time course of glutamate at the synapse and post synaptic glutamate receptor activation [68,69]. Glutamate transporters are present in astrocytic processes that sheathe synapses, and their proximity to the synapse is mediated by dynamic rearrangements of astrocytic processes that influence their ability to clear extracellular glutamate [70,71].

Work by Ameroso et al. indicates that elevated levels of BDNF in the fed state activate astrocytic TrkB.T1 receptors in the VMH, leading to the retraction of astrocytic processes neighboring excitatory synapses and inhibition of GLT-1 expression [64]. These caloric status/BDNF-driven anatomical and molecular changes in VMH astrocytes are proposed to restrict synaptic glutamate clearance, thereby enhancing excitability and activity of VMH neurons and increasing the anorexigenic tone and energy expenditure. In support, selective TrkB.T1 knockdown in VMH astrocytes (TrkB.T1 KD) in mice induced hyperphagia, obesity, and glucose intolerance as well as reduced locomotor activity and thermogenesis [64]. TrkB.T1 KD mutants displayed blunted responses to leptin treatment, indicating that astrocytic TrkB.T1 in the VMH is required for the effects of leptin of decreasing food intake and body weight. These metabolic alterations were associated with increased expression of GLT-1 and the proximity of astrocytic processes to excitatory synapses, which enhanced synaptic glutamate clearance and decreased neuronal excitability and activity in the VMH. Of note, when *Ntrk2* was targeted in VMH neurons, glycemic control was impaired, whereas feeding and body weight were unaffected, indicating differential effects of BDNF signaling in neurons versus astrocytes [64]. The cumulative data indicate that BDNF signaling in astrocytes is requisite for energy status-driven excitatory synaptic plasticity in the VMH and energy balance control.

The role of BDNF regulating GABAergic inhibitory transmission in the VMH in a way that influences metabolic function is less clear. One study by the Simerly group indicates that BDNF in the VMH is a key signal in the organization of GABAergic circuits in this region [61]. Accordingly, deletion of *Bdnf* in SF1 neurons induced an enduring elevation in inhibitory synapse density in the VMH, as indicated by a 21.7% increase in vesicular gaba trasnporter (VGAT) immoreactive puncta in contact with SF1 neurons. In contrast, the number of excitatory inputs, as marked by vGlut2 immunoreactivity onto SF1 neurons, was unchanged in mutant mice. As stated earlier, SF1^BDNF^ mutant mice have a deficient counterregulatory response to hypoglycemia [61]. The data suggest that the elevated inhibitory control onto SF1 neurons elicited by BDNF depletion might underlie this metabolic dysfunction. This is consistent with key roles in glycemic control previously ascribed to this cell population [55].

The evidence outlined thus far establishes that BDNF signaling plays chief roles in the synaptic plasticity and function of metabolic circuits in the VMH. It is also important to understand the underlying molecular mechanisms and how blunted BDNF signaling in the VMH might corrupt them, leading to metabolic dysfunction. α2δ-1, which colocalizes with TrkB-expressing cells in the VMH [72], has emerged as a candidate downstream effector of BDNF. α2δ-1 is expressed in neurons and functions as an auxiliary high-voltage calcium channel subunit that facilitates cell surface channel expression, thereby increasing calcium currents and neurotransmitter release [73,74]. It also promotes excitatory synaptogenesis in a calcium channel-independent manner by acting as a receptor for astrocyte-derived thrombospondins (TSP) [75]. We found that deficits in the cell surface expression of α2δ-1 in mice with central depletion of BDNF (BDNF^2L/2LCk-Cre^) contribute significantly to the disrupted energy balance control of these animals. In support, viral-mediated rescue of deficient α2δ-1 expression in the VMH of BDNF^2L/2LCk-cre^ mice mitigated but did not normalize their hyperphagia and obesity without affecting locomotor activity. Remarkably, circulating levels of glucose and responses to a glucose challenge were normalized and liver steatosis was dramatically improved in rescued BDNF mutants even though these animals remained 42% heavier than controls. These findings suggest that α2δ-1 is a downstream effector of BDNF that governs mechanisms in this region that can uncouple glucose balance and liver fat deposition from obesity. Furthermore, because VMH neurons in BDNF^2L/2LCk-cre^ mutants display normal calcium currents but significantly decreased density of excitatory synapses and frequency of sEPSCs [64,72], it is likely that effects of α2δ-1 are related to its role as a TSP receptor and not an auxiliary calcium channel subunit. Further work is needed to determine how central BDNF deletion results in the reduction in cell surface levels of α2δ-1. This line of research warrants examination considering that weight gain is a noted contraindication for gabapentinoid drugs used to treat neuropathic pain and seizure disorders and that are known to inhibit α2δ-1 [76,77,78].

mGluR5 signaling is another putative downstream molecular mechanism mediating the effects of BDNF in the VMH. In support, male and female BDNF^2L/2LCk-cre^ mutant mice show a significant reduction in mGluR5 expression in the VMH. Furthermore, mGluR5 expression in the VMH is reduced in the fasted state, similar to the expression of BDNF and TrkB [79]. mGluR5 is highly expressed in VMH neurons and in astrocytes associated with excitatory synapses in this region [80]. Developmental mGluR5 depletion in SF1 neurons or in VMH astrocytes in adulthood does not alter food intake, body weight, or locomotor activity in males or females [79,81]. These findings suggest that mGluR5 is not an essential mediator of actions of BDNF in the VMH facilitating energy homeostasis. In contrast, neuronal or astrocytic depletion of mGluR5 in VMH significantly affected glucose and lipid homeostasis. Albeit not having any effect on body weight, mGluR5 depletion in SF1 neurons resulted in impaired insulin and glucose tolerance, increased WAT adiposity, and decreased sympathetic output in females but not in males [79]. These metabolic dysfunctions in female mutants were related to reduced intrinsic excitability as well as reduced and increased excitatory and inhibitory drive, respectively, onto SF1 neurons [79]. In contrast, astrocytic mGluR5 depletion in the adult VMH resulted in enhanced glucose tolerance and increased glucose-induced insulin secretion in both males and females [81]. These changes in glycemic control were linked to reduced excitatory drive onto pituitary adenylate cyclase-activating polypeptide (PACAP) neurons in this region, which normally inhibit insulin secretion [81]. Depletion of astrocytic mGluR5 in the VMH also reduced adipocyte size in WAT, and this effect was accompanied by elevations in norepinephrine content specifically in gWAT, indicating upregulated sympathetic tone in this tissue. The results indicate complex context-dependent effects of mGluR5 impacting glucose and lipid homeostasis.

The cumulative data indicate that BDNF signaling in neurons and astrocytes in the adult VMH facilitates caloric state-driven plasticity and activity of local circuits influencing energy, glucose, and lipid balance. Even though perturbing BDNF signaling in the VMH does not result in severe obesity, as observed in mice depleted of BDNF or TrkB in the PVN, it significantly compromises feeding and weight control, with concomitant alterations in glucose homeostasis. Future investigations should aim at delineating the relevant input and output connections of VMH BDNF^+^ neurons mediating these effects.

### 2.4. Dorsomedial Hypothalamus

The DMH houses both orexigenic and anorexigenic cell populations, including neurons sensitive to leptin that suppress feeding and increase energy expenditure via elevated thermogenesis and locomotion [82,83,84]. It has reciprocal connections with several hypothalamic nuclei, including projections to the hindbrain [85]. A role for BDNF signaling in the DMH mediating metabolic health is indicated by studies examining the effect of deleting *Ntrk2* in this region in adult mice. Accordingly, mice with neuronal TrkB knock down in adult DMH exhibited obesity, hyperphagia, impaired glucose tolerance, and decreased energy expenditure [19]. TrkB-containing neurons in the DMH are distinct from those expressing leptin receptors, and unlike those cells, while projecting to the Arc they do not synapse onto AgRP or POMC neurons in that region [19,86]. They comprise glutamatergic and GABAergic subpopulations located in the central and lateral DMH or central and ventral DMH, respectively. Interestingly, both subpopulations respond with increased activity during refeeding following an overnight fast in mice. Chemogenetic activation of DMH^TrkB^ neurons during the dark cycle, when energy intake is greater in mice, is sufficient to suppress appetite, as indicated by an 82% reduction in food intake [19]. In contrast, selective inhibition of DMH^TrkB^ neurons during the dark cycle did not affect food consumption, suggesting that these cells are silenced at that time of the day when animals are physiologically hungry and need to replenish energy stores. However, inhibition of DMH^TrkB^ neurons during the light cycle significantly elevated food intake, indicating that these cells restrict feeding during that time period.

DMH^TrkB^ neurons also play chief roles regulating thermogenesis and energy expenditure. In support, TrkB^+^ neurons in the DMH are activated by both cold and warm ambient temperatures [86]. Moreover, selective chemogenetic activation of DMH^TrkB^ neurons robustly increases adaptive thermogenesis, locomotor activity, and energy expenditure without affecting heart rate or blood pressure. Studies involving activation of projection-defined DMH^TrkB^ neurons revealed that projections of these cells to the raphe pallidus elevate thermogenesis and energy expenditure, whereas those sending collaterals to the PVN and preoptic area inhibit feeding [86].

The role of DMH^BDNF^ neurons has also been investigated using chemogenetic approaches. A recent report indicates that DMH^BDNF^ neurons are activated by afferent cooling signals, with concomitant increases in body temperature, energy expenditure, and locomotor activity [87]. Conversely, selective inhibition of DMH^BDNF^ neurons reduces energy expenditure, locomotor activity, and UCP1 expression in BAT, resulting in increased body weight. In total, the results indicate an important role of BDNF and TrkB-containing neurons in the DMH in mediating thermoregulation as well as feeding and energy expenditure control. Future studies should aim to interrogate and contrast the roles of the glutamatergic and GABAergic TrkB^+^ subpopulations in the DMH.

### 2.5. Lateral Hypothalamus

The lateral hypothalamus is frequently described as a feeding center, considering that early studies showed that lesions in this area elicit hypophagia [88]. It harbors cells that produce melanin-concentrating hormone (MCH) and hypocretin, both recognized as orexigenic factors that stimulate appetite [89,90]. Although BDNF^+^ and TrkB^+^ cells are present in the LH [8,20], their roles there remain poorly understood. One study showed that selective disruption of *Bdnf* expression from promoter 1 (Bdnf-e1), which is highly expressed in this region, led to significant obesity, decreased sympathetic activity within BAT, and impaired adaptive thermogenesis [91]. These deficits in thermogenic control were associated with reduced BAT expression of Ucp1 and Pcg1a, genes critical for regulating thermogenesis. Notably, Bdnf-e1-expressing neurons in the LH are polysynaptically connected to BAT, and delivery of a TrkB agonist antibody to the LH of Bdnf-e1 mutants reversed the alterations in body temperature and UCP1 expression [91]. The data indicate that BDNF expression in the LH plays a key role in the regulation of thermogenesis.

### 2.6. Preoptic Area

The preoptic area (POA) of the hypothalamus plays an important role in regulating body temperature, as illustrated by early studies showing that lesions to this region impair thermoregulatory responses following temperature challenges [92]. It receives considerable sensory input from the skin, informing environmental temperatures, and both warm- and cold-responsive neurons can be found in the POA thermoregulatory circuitry [93]. A link between BDNF in the POA and body temperature regulation was first suggested by work showing that BDNF mRNA expression there was elevated during both heat and cold exposure in 3-day-old chicks [94]. Furthermore, knocking down BDNF expression in this region during development impaired thermal regulation of the chicks later in life. The findings indicate that BDNF plays an essential role in the development of circuits in the POA regulating thermal control.

In contrast to TrkB^+^ neurons in the DMH, which are activated by both cold and warm temperatures [86], fiber photometry studies in mice showed that cells co-expressing BDNF and PACAP in the POA were selectively activated by environmental warmth but not by a cold challenge. These warm sensitive neurons project to the DMH, and their optogenetic activation rapidly elicits hypothermia [95]. These effects are associated with autonomic and behavioral responses that reduce thermogenesis and increase cold-seeking behavior. The findings indicate that PACAP/BDNF^+^ neurons in the POA coordinate autonomic and behavioral responses to environmental warmth to achieve temperature homeostasis.

## 3. Extrahypothalamic Actions of BDNF and TrkB Influencing Feeding and Body Weight

### 3.1. Dorsal Vagal Complex

The dorsal vagal complex (DVC) is located in the caudal brainstem and comprises several nuclei playing a critical role in the regulation of energy balance, including the area postrema (AP), the nucleus tractus solitarious (NTS), and the dorsal motor nucleus of the vagus (DMV). It receives several modalities of information via vagal sensory neurons in the gut, including levels of gastric distention and the presence of appetite-regulating peptides and nutrients in the gut, which triggers the satiety reflex, terminating each meal as the short-term control of feeding [96,97,98]. Moreover, bloodborne factors informing the caloric status of the animal enter the AP, which is classified as a circumventricular organ with a permissive blood–brain barrier. The DVC is primarily involved with the suppression of appetite via homeostatic and aversive mechanisms. These effects are mediated at least in part through connections with the hypothalamus and the parabrachial nucleus. It also has a high density of MC4R and is an important site of melanocortin signaling.

Whereas BDNF-containing cell bodies and fibers are present in the NTS, numerous TrkB^+^ cells are located in the NTS and area postrema [99,100]. Consistent with a role in energy balance control, BDNF expression in the DVC is sensitive to energy status and increased in the fed state. Furthermore, exogenous administration of the anorexigenic factors leptin, cholecystokinin (CCK), and MC3R/MC4R agonists elevate BDNF levels in this region [101,102,103]. Evidence of a functional relationship between BDNF in the DVC and appetite control includes the finding that infusion of exogenous BDNF into this area reduces food intake and body weight in rodents [101]. Moreover, pharmacological blockade of TrkB abrogated the anorexigenic effects of a selective MC4R agonist delivered to the DVC, suggesting that facilitating the actions of melanocortin signaling in this region is a candidate mechanism underlying the effects of BDNF [102]. Another pharmacological study in rats showed that infusion of BDNF into the NTS potently suppressed food intake and resulted in weight loss [103]. Notably, whereas leptin administration to the NTS reduced food intake in rats, this anorexigenic effect was abolished when the TrkB receptor antagonist ANA-12 was delivered to the NTS prior to leptin treatment, indicating that BDNF/TrkB signaling facilitates the effects of leptin in the NTS.

Genetic studies further suggest the key role of BDNF signaling in the DVC facilitating energy homeostasis. Accordingly, mice with *Ntrk2* deletion in the hindbrain (Phox2b-Ntrk2+/−) are hyperphagic but surprisingly exhibit no effects on body weight, adiposity, or glucose homeostasis. It is possible that alterations in energy expenditure counteract the effects of excessive feeding, but this parameter was not measured in these studies [104]. Because BDNF/TrkB signaling in the DVC has been under studied relative to the hypothalamus, much remains to be understood regarding its roles there, the underlying mechanisms, and the relevant input and output connections.

### 3.2. Dopaminergic Reward Circuits

Homeostatic mechanisms balancing nutritional requirements and caloric status operate primarily, although not exclusively, within the hypothalamus and hindbrain. Hedonic feeding is mediated by reward circuits regulating motivated behaviors that can override homeostatic cues to drive consumption of highly palatable food. These homeostatic and hedonic systems are not completely disassociated but instead are anatomically connected and can cooperate to orchestrate feeding responses. The dopamine (DA) circuitry plays key roles in reward-driven food intake [105,106,107]. This includes the mesolimbic dopamine pathway comprising DA neurons in the ventral tegmental area (VTA) that target the nucleus accumbens (NAc), the medial prefrontal cortex (mPFC), and the amygdala.

BDNF is expressed in DA neurons residing in the VTA as well as in neurons in the mPFC, from where it is anterogradely transported to the NAc, which exhibits little to no expression of BDNF [108,109]. TrkB is produced in VTA DA neurons, PFC, and GABAergic medium spiny-projection neurons in the NAc [108,109,110]. Consumption of palatable HFD influences expression of both BDNF and TrkB in the VTA [111]. Notably, amperometric recordings in acute slices showed that female BDNF^2L/2LCk-cre^ mutants with global brain depletion of BDNF display a reduction in evoked release of dopamine in the NAc shell but not the NAc core. This deficit persisted following addition of the dopamine transporter (DAT) inhibitor nomifensine, indicating deficient presynaptic release of dopamine in the NAc in the absence of BDNF [111]. The mechanisms leading to reduced mesolimbic dopamine secretion in BDNF mutants are not yet fully understood. It is possible that BDNF plays a role in modulating the excitability of VTA DA neurons during processes related to food reward. Supporting this notion, Pu et al. demonstrated that BDNF is essential to the enhancement of excitatory synapses on VTA dopamine neurons after cocaine withdrawal in rat brain slices [112].

To determine whether observed perturbations in the mesolimbic dopaminergic system might contribute to the hyperphagic behavior displayed by BDNF^2L/2LCk-cre^ mutant mice, mice with selective targeting of *Bdnf* in the adult VTA were examined. VTA^BDNF^ deficient mice exhibited significant increases in food intake and body weight when fed a HFD compared to controls [111]. However, VTA^BDNF^ mutants displayed no significant differences in caloric intake or body weight when fed a chow diet. These data indicate that BDNF is a key regulator of the mesolimbic dopamine system and that actions there critically regulate hedonic but not homeostatic feeding.

### 3.3. Amygdala

The amygdala, a component of the limbic system, is an integrative center for emotional processing, including fear and anxiety [113]. Although it is now evident that circuits involving this brain region influence body weight, the underlying mechanisms are far less clear. The amygdala has reciprocal connections with the hypothalamus, and reward systems influencing hedonic feeding and BDNF and TrkB are expressed in this brain area [114]. Studies showed that adolescent female rats that underwent the activity-based anorexia (ABA) protocol had decreased TrkB signaling in the amygdala albeit elevated content of BDNF protein, and these effects persisted after a period of body-weight recovery [115]. The ABA protocol involves caloric restriction combined with increased exercise in running wheels and paradoxically elicits reductions in food intake and severe weight loss [116]. Consistent with diminished BDNF signaling, the phosphorylation of TrkB in Tyr(Y)706 and its downstream targets were reduced in amygdala from ABA rats. These findings suggest a link between diminished BDNF signaling in the amygdala and changes in feeding behavior following ABA.

Genetic studies further support the role of BDNF in the amygdala influencing body weight. Anxiety is often associated with weight loss, but the underlying mechanisms are incompletely understood. Xie et al. found that BDNF depletion in the hippocampus, cortex, and amygdala in mice (Bdnf^lox/lox^;Emx1-cre) resulted in impaired GABAergic transmission accompanied by elevated levels of anxiety-like behavior and resistance to diet-induced obesity [117]. These mutants also exhibited increases in circulating levels of the stress hormone corticosterone as well as elevated peripheral sympathetic tone, basal metabolic rate, and thermogenesis. Follow-up studies indicated that the amygdala is the critical source of BDNF linking anxiety behaviors and metabolic function. Firstly, BDNF re-expression in the amygdala of Bdnf^lox/lox^;Emx1-cre mutant mice reversed the previously observed alterations in anxiety-related behavior and metabolic function. Additionally, amygdala-specific BDNF knockdown similarly produced anxiety-like behavior, increased energy expenditure, and conferred protection against diet-induced obesity [117]. These findings suggest that deficits in BDNF function in the amygdala lead to the activation of anxiogenic circuits, which in turn increase sympathetic outflow with concomitant increases in thermogenesis and energy expenditure. This work illustrates a mechanism governed by BDNF and linking anxiety to changes in energy metabolism. This is in contrast to the effects of depleting BDNF in the hypothalamus described earlier [21,36], which resulted in reduced energy expenditure and increased body weight and food consumption and thus exemplifies the complex and context-dependent effects of this neurotrophin.

## 4. Summary

The effects of BDNF and TrkB on metabolic function are multifaceted, complex, and context-dependent (Table 1). Compelling evidence attests to their prominent and essential roles in the regulation of homeostatic and hedonic feeding as well as in the regulation of energy expenditure through mechanisms acting in the hypothalamus, the DVC, and the mesolimbic dopamine reward pathway. Acting in the amygdala, BDNF provides a mechanistic link between alterations in metabolic function associated with anxiety. Acknowledging that metabolic circuits are not hardwired but rather are plastic and adapt to meet the energy demands of the animal, it should not be surprising that BDNF, a chief regulator of synaptic plasticity, plays such paramount roles in the adult brain. Considering its established functions in facilitating structural plasticity within the hippocampus and cortex [118,119], it is possible that BDNF guides the reorganization of feeding circuits in response to metabolic signals, thereby promoting energy balance. Work in the VMH showing that neuronal and astrocytic components of metabolic excitatory circuits there adapt to changes in caloric status in a BDNF-dependent manner indicates that this is indeed the case [64]. Because BDNF plays a crucial role in regulating local protein synthesis at synapses [120], it could also facilitate swift post-meal alterations in the synaptic proteome of neurons that regulate feeding, energy expenditure, and glucose mobilization. The reported sexually dimorphic effects of perturbing BDNF signaling should also be more carefully explored. Considering the significant prevalence of obesity and mutations affecting BDNF signaling within the human population [121], elucidating the mechanistic outcomes of BDNF/TrkB signaling affecting energy and glucose balance is essential. It is also important to consider that BDNF and TrkB might not be optimal targets for intervention strategies for metabolic disorders, as they are broadly expressed in the brain and play a multitude of functions. Therefore, defining downstream pathways mediating their actions would be valuable in devising new treatment approaches for obesity and its numerous related health issues.

## Figures and Tables

**Figure 1 biomolecules-14-00424-f001:**
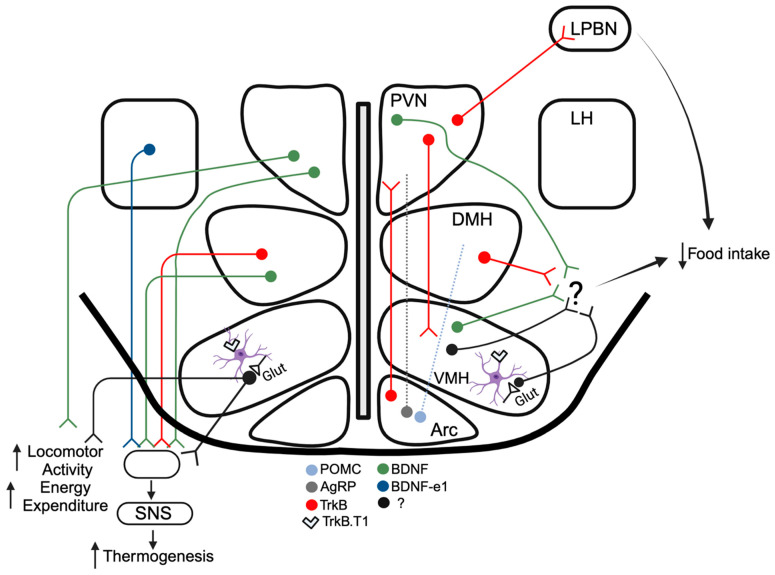
Roles of BDNF and TrkB in the hypothalamus influencing energy homeostasis. Whereas the right side of the diagram illustrates roles affecting food intake, the left side shows functions impacting thermogenesis, locomotor activity, and energy expenditure. Cell types are color coded as indicated. Dashed lines represent the effects of BDNF signaling-mediating development of POMC and AgRP axons targeting the PVN and DMH. Regulation of astrocytic (purple cell) synaptic glutamate clearance under the control of BDNF/TrkB.T1 is shown in the VMH. Cells involved in thermoregulation are represented as being part of polysynaptic circuits regulating sympathetic nervous system (SNS) output. Arc, arcuate nucleus; VMH, ventromedial hypothalamus; DMH, dorsomedial hypothalamus; PVN, paraventricular nucleus; LH, lateral hypothalamus; LPBN, lateral parabrachial nucleus; Glut, glutamatergic synapse. This figure was produced using BioRender.

**Table 1 biomolecules-14-00424-t001:** Roles of BDNF and TrkB in the regulation of energy and glucose balance in the hypothalamus and extrahypothalamic tissues.

Brain Region	Food Intake	Thermogenesis	Locomotor Activity	Energy Expenditure (EE)	Glucose Balance
Arcuate nucleus (Arc)	↑ Activation of TrkB^+^ neurons during refeeding following fasting period [35]				
Pariventricular nucleus (PVN)	Depletion of BDNF or TrkB ↑ food intake [36,52]Activation of BDNF^+^ or TrkB^+^ neurons ↓ food intake [48,52]	Depletion of BDNF ↓ thermogenesis [36] Activation of BDNF^+^ neurons ↑ thermogenesis [48]	Depletion of BDNF or TrkB ↓ locomotor activity [36,52]Activation of BDNF^+^ or TrkB^+^ neurons ↑ locomotor activity [48,52]	Depletion of BDNF ↓ EE [36]Activation of BDNF^+^ neurons ↑ EE in females [48]	Depletion of BDNF results in glucose intolerance and hyperinsulinemia [36]
Ventromedial hypothalamus (VMH)	Depletion of BDNF or Bdnf-e2 ↑ food intake [21,63] Depletion of Bdnf-e1 ↑ food intake only when socially isolated or on HFD [63]Depletion of astrocytic TrkB.T1 ↑ food intake [64]	Depletion of astrocytic TrkB.T1 ↓ thermogenesis [64]	Depletion of astrocytic TrkB.T1 ↓ locomotor activity [64]		BDNF depletion induces hyperglycemia and hyperinsulinemia [21]Depletion of astrocytic TrkB.T1 results in glucose intolerance [64]
Dorsomedial hypothalamus (DMH)	Activation of TrkB^+^ neurons ↓ food intake, whereas inhibition of TrkB^+^ neurons ↑ food intake [19]	Activation of BDNF^+^ neurons in response to cold temperatures ↑ thermogenesis [87]Activation of TrkB^+^ neurons ↑ adaptive thermogenesis [86]	Activation of BDNF^+^ neurons ↑ locomotor activity [87] Activation of TrkB^+^ neurons ↑ locomotor activity [86]	Activation of BDNF^+^ neurons ↑ EE [87]Activation of TrkB^+^ neurons ↑ EE [86]	Depletion of TrkB induces glucose intolerance [19]
Lateral hypothalamus (LH)		Depletion of Bdnf-e1 ↓ adaptive thermogenesis [91]TrkB agonist ↑ thermogenesis [91]			
Preoptic area (POA)		Activation of BDNF/PACAP^+^ neurons by environmental warmth ↓ thermogenesis [95]			
Dorsal vagal complex (DVC)	Infusion of BDNF into DVC ↓ food intake [103]Depletion of TrkB^+^ ↑ food intake [104]				
Ventral tegmental area (VTA)	Knock down of BDNF ↑ HFD but not chow intake [111]				
Amygdala	Depletion of BDNF elicits resistance to DIO [117]	Depletion of BDNF ↑ thermogenesis [117]		Depletion of BDNF ↑EE [117]

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
