# Peer review of "The Role of BDNF and TrkB in the Central Control of Energy and Glucose Balance: An Update"

_biomolecules, 2024, doi:10.3390/biom14040424_

Round 1
Reviewer 1 Report
Comments and Suggestions for Authors
This manuscript is a splendid updated review of the puzzling anorexigenic function of the neurotrophin BDNF in mammalian brain, in a synthetic and exhaustive survey from organism to molecular levels. The review is very well structured and written. There are just a few small details to correct so as to render it ready to publish. Congratulations to the authors!
Scientific issues
Lines 485-503 : ?? POA and BDNF and reproduction control?
L. 511: Add “which triggers the satiety reflex, terminating each meal as the short-term feedback control of feeding” ahead of [96-98].
Minor details
Line 208 : Change « involvent » into « involvement ».
L. 231-238: Is there any reason for switching to italic mode here?
L.387: Change “normalized” into “normalize”.
L. 485: Change “Coversely” into “Conversely”
L. 533: Change “facilating” into “facilitating”.
Author Response
We thank the reviewer for their comments. We have addressed issues raised as follows:
Lines 485-503 : ?? POA and BDNF and reproduction control?
Please note that in this review we focus solely on actions of BDNF and TrkB impacting feeding, body weight, as well as energy and glucose balance control.
- 511: Add “which triggers the satiety reflex, terminating each meal as the short-term feedback control of feeding” ahead of [96-98].
Please note this statement has been added.
The following edits have been made:
Line 208 : Change « involvent » into « involvement ».
- 231-238: Is there any reason for switching to italic mode here?
L.387: Change “normalized” into “normalize”.
- 485: Change “Coversely” into “Conversely”
- 533: Change “facilating” into “facilitating”.
.
Reviewer 2 Report
Comments and Suggestions for Authors
This nicely written review report provides a brief insight into the role of central substrate BDNF and TrkB receptors in regulating energy homeostasis. A few comments to enhance the readership of the review:
1) The author should add a table highlighting the brain regions (including hypothalamic or extrahypothalamic regions) where BDNF/TrkB receptor positively or negatively regulates energy homeostasis or glucose metabolism.
2) VMH where both BDNF and TrkB are expressed whether there is positive feedback look exists needs to be clarified.
3) Given that BDNF and its receptor play a crucial role in energy balance what progress so far has been made in the drug development to prevent obesity.
Author Response
We thank the reviewer for their consideration of our review article. We have addressed the issues raised as follows:
1) The author should add a table highlighting the brain regions (including hypothalamic or extrahypothalamic regions) where BDNF/TrkB receptor positively or negatively regulates energy homeostasis or glucose metabolism.
Please note that a table is now included in the revised manuscript containing this information.
2) VMH where both BDNF and TrkB are expressed whether there is positive feedback look exists needs to be clarified.
We now describe work in the VMH indicating that BDNF and TrkB co-localize in this region suggesting autocrine actions impacting metabolic control
3) Given that BDNF and its receptor play a crucial role in energy balance what progress so far has been made in the drug development to prevent obesity.
To our knowledge, this pathway is not being considered for drug development to treat obesity. This wouldn't be surprising considering that TrkB is broadly expressed in the brain and BDNF has such a multitude of roles in the CNS, raising the risk for undesired contraindications. We added a statement refereeing to this in the Summary section of the manuscript.